

# Statistical properties of a stochastic model of eddy hopping

Izumi Saito[1], Takeshi Watanabe[1], and Toshiyuki Gotoh[1]

[1]Department of Physical Science and Engineering, Nagoya Institute of Technology, Gokiso-cho, Showa-ku, Nagoya 466-8555, Japan

**Correspondence:** Izumi Saito (izumi@gfd-dennou.org)

**Abstract.**

Statistical properties are investigated for the stochastic model of eddy hopping, which is a novel cloud microphysical model that accounts for the effect of the supersaturation fluctuation at unresolved scales on the growth of cloud droplets and on spectral broadening. It is shown that the model fails to reproduce a proper scaling for a certain range of parameters, resulting in

a deviation of the model prediction from the reference data taken from direct numerical simulations and large-eddy simulations (LESs). Corrections to the model are introduced so that the corrected model can accurately reproduce the reference data with the proper scaling. In addition, a possible simplification of the model is discussed, which may contribute to a reduction in computational cost while keeping the statistical properties almost unchanged in the typical parameter range for the model implementation in the LES Lagrangian cloud model.

## 1   Introduction

The purpose of the present paper is to investigate the statistical properties of the stochastic model of eddy hopping proposed by Grabowski and Abade (2017). This stochastic model, referred to hereinafter as the eddy hopping model, was developed in order to account for the effect of the supersaturation fluctuation at unresolved (subgrid) scales on the growth of cloud droplets by the condensation process. In a turbulent cloud, cloud droplets arriving at a given location follow different trajectories and

thus experience different growth histories, which leads to significant spectral broadening. This mechanism, referred to as the stochastic condensation theory, has been investigated since the early 1960s by a number of researchers [mostly Russian, see Sedunov (1974); Clark and Hall (1979); Korolev and Mazin (2003)], but the importance of this mechanism was later reinforced by Cooper (1989); Lasher-Trapp et al. (2005). For this mechanism, Grabowski and Wang (2013) emphasized the importance of large-scale eddies (turbulent eddies with scales not much smaller than the cloud itself) and proposed the concept of large-eddy

hopping. Grabowski and Abade (2017) formulated this concept and developed the eddy-hopping model. For the following studies using the eddy-hopping model, readers are referred to Abade et al. (2018); Thomas et al. (2020).

In the present paper, we take a rather theoretical approach to obtain various statistical properties of the eddy-hopping model, such as the variance, covariance, and auto-correlation function of the supersaturation fluctuation. We show that the model fails to reproduce a proper scaling for a certain range of parameters, resulting in the deviation of the model prediction from the

reference data taken from direct numerical simulations (DNSs) and large-eddy simulations (LESs). We introduce corrections to the model so that the corrected model can accurately reproduce the reference data with the proper scaling. We also discuss





the possibility of simplification of the model, which may contribute to a reduction in computational cost while keeping the statistical properties almost unchanged in the typical parameter range for the model implementation in the LES Lagrangian cloud model.

The remainder of the present paper is organized as follows. Section 2 describes the governing equations. Section 3 presents a theoretical analysis and numerical experiments and demonstrates the improper scaling in the model prediction. Section 4 introduces corrections to the model. Finally, Section 5 discusses the possibility of simplification of the model.

## 2    Governing equations

The eddy-hopping model proposed by Grabowski and Abade (2017) consists of the following two evolution equations. First,
the fluctuation of the vertical velocity of turbulent flow at the droplet position, $w'(t)$, is modeled by the Ornstein-Uhlenbeck process:

$$w'(t+\delta t) = w'(t)e^{-\delta t/\tau} + \sqrt{1 - e^{-\frac{2\delta t}{\tau}}} \, \sigma_{w'}\psi, \tag{1}$$

where $\delta t$ is the time increment, $\psi$ is a Gaussian random number with zero mean and unit variance drawn every time step, $\sigma_{w'}$ is the standard deviation of $w'$, and $\tau$ is the integral time, or the large-eddy turnover time of the turbulent flow. Here, $\sigma_{w'}$ and
$\tau$ are used as the model parameters. Second, the supersaturation fluctuation at the droplet position, $S'(t)$, is governed by

$$\frac{dS'}{dt} = a_1 w' - \frac{S'}{\tau_{relax}}. \tag{2}$$

Here, the first term on the right-hand side represents the effect of adiabatic cooling/warming due to air parcel ascent/descent caused by the vertical velocity $w'(t)$. The parameter $a_1$ has the unit of a scalar gradient. The second term on the right-hand side represents the effect of condensation/evaporation of droplets. The time scale $\tau_{relax}$ is referred to as the phase relaxation time
and is inversely proportional to the average of the number density and radius of the droplets (Politovich and Cooper, 1988; Korolev and Mazin, 2003; Kostinski, 2009; Devenish et al., 2012).

Equation (1) can also be written as the following derivative form (Pope, 2000):

$$\frac{dw'}{dt} = -\frac{1}{\tau}w'(t) + F_{w'}(t). \tag{3}$$

Here, the term $F_{w'}(t)$ is statistically independent of $S'$ and obeys the Gaussian random process that has zero mean and two-time
covariance defined by

$$\langle F_{w'}(t)F_{w'}(s)\rangle = \left(\frac{2\sigma_{w'}^2}{\tau}\right)\delta(t-s), \tag{4}$$

where the angle brackets indicate an ensemble average and $\delta(\,)$ is the Dirac delta function. In the following theoretical analysis, Eqs. (3) and (2) are used as the governing equations of the eddy-hopping model.



## 3   Statistical properties of the model

We now obtain the analytical expression for the standard deviation of the supersaturation fluctuation, $\sigma_{S'}$, in a statistically steady state. Starting from Eqs. (3) and (2), the result is provided in Eq. (13).

First, multiplying Eq. (3) by $S'$ and taking an ensemble average, we obtain

$$\left\langle S' \frac{dw'}{dt} \right\rangle = -\frac{1}{\tau} \langle w' S' \rangle \tag{5}$$

because of statistical independence ($\langle S' F_{w'} \rangle = 0$). Second, multiplying Eq. (2) by $w'$ and taking an ensemble average, we

obtain

$$\left\langle w' \frac{dS'}{dt} \right\rangle = a_1 \langle w'^2 \rangle - \frac{1}{\tau_{relax}} \langle w' S' \rangle. \tag{6}$$

Summing Eqs. (5) and (6), we obtain

$$\frac{d}{dt} \langle w' S' \rangle = a_1 \langle w'^2 \rangle - \frac{1}{\tau_{relax}} \langle w' S' \rangle - \frac{1}{\tau} \langle w' S' \rangle. \tag{7}$$

Next, we consider a statistically steady state. Since an ensemble-averaged variable does not change in time ($d \langle \circ \rangle / dt = 0$) and

$\langle w'^2 \rangle = \sigma_{w'}^2$ in the statistically steady state, we obtain the flux of the supersaturation in the vertical direction as follows:

$$
\begin{aligned}
\langle w' S' \rangle &= a_1 \left( \frac{1}{\tau} + \frac{1}{\tau_{relax}} \right)^{-1} \sigma_{w'}^2 \\
&= a_1 (1 + Da)^{-1} \tau \sigma_{w'}^2,
\end{aligned}
\tag{8}
$$

where $Da$ is the Damköhler number (Shaw, 2003) defined as

$$Da = \frac{\tau}{\tau_{relax}}. \tag{9}$$

Next, multiplying Eq. (2) by $S'$ and taking an ensemble average, we obtain

$$\frac{1}{2} \frac{d}{dt} \langle S'^2 \rangle = a_1 \langle w' S' \rangle - \frac{1}{\tau_{relax}} \langle S'^2 \rangle. \tag{10}$$

In the statistically steady state, we have

$$\sigma_{S'}^2 = \langle S'^2 \rangle = a_1 \tau_{relax} \langle w' S' \rangle. \tag{11}$$

Combining Eqs. (8) and (11), we obtain

$$
\begin{aligned}
\sigma_{S'}^2 &= a_1 \tau_{relax} \left[ a_1 (1 + Da)^{-1} \tau \sigma_{w'}^2 \right] \\
&= a_1^2 (1 + Da)^{-1} \tau_{relax} \tau \sigma_{w'}^2,
\end{aligned}
\tag{12}
$$

or equivalently,

$$\sigma_{S'} = (1 + Da)^{-1/2} Da^{-1/2} a_1 \tau \sigma_{w'}. \tag{13}$$

Here, $\sigma_{S'}$ in Eq. (13) has two important asymptotic forms, as shown below:





1. *Large scale limit*

For $\tau \to \infty$ (or equivalently, $Da \to \infty$, $L \to \infty$, where $L = \sigma_{w'}\tau$ is the integral scale), $\sigma_{S'}$ in Eq. (13) is approximated as

$$
\begin{aligned}
\sigma_{S'} &\approx a_1 Da^{-1/2}\tau_{relax}^{1/2}\tau^{1/2}\sigma_{w'} \\
&= a_1\tau_{relax}\sigma_{w'}.
\end{aligned} \tag{14}
$$

For the case of a constant dissipation rate of turbulent kinetic energy $\varepsilon$, $\sigma_{w'} \sim L^{1/3}$ (see Appendix B), and we have the following scaling:

$$
\sigma_{S'} \sim L^{1/3}. \tag{15}
$$

    2. *Small scale limit*

For $\tau \to 0$ (or equivalently, $Da \to 0$, $L \to 0$), $\sigma_{S'}$ in Eq. (13) is approximated as

$$
\sigma_{S'} \approx a_1\tau_{relax}^{1/2}\tau^{1/2}\sigma_{w'}. \tag{16}
$$

For the case of a constant dissipation rate of turbulent kinetic energy $\varepsilon$, $\sigma_{w'} \sim L^{1/3}$ and $\tau \sim L^{2/3}$ (see Appendix B), and we have the following scaling:

$$
\sigma_{S'} \sim L^{2/3}. \tag{17}
$$

The above asymptotic forms of $\sigma_{S'}$ in the two limits can be validated through comparison with the result of the scaling argument

by Lanotte et al. (2009). From their argument, we should have $\sigma_{S'} \sim a_1\tau_{relax}\sigma_{w'}$ for the large scale limit, which is consistent with Eq. (14). On the other hand, we should have $\sigma_{S'} \sim a_1\tau\sigma_{w'}$ for the small scale limit, which is inconsistent with Eq. (16). Therefore, the eddy-hopping model given by Eqs. (3) and (2) does not reproduce the proper scaling for the small scale limit.

    Figure 1 compares the scale dependence of $\sigma_{S'}$ for the analytical expression given by Eq. (13) (orange curve) with the results of the numerical integration of the eddy-hopping model given by Eqs. (1) and (2) (blue squares). Here, numerical integration

is conducted in the same manner as that by Thomas et al. (2020) (Section 5 of their study), except that the integration time is increased from $6\tau$ to $10\tau$ (see Appendix A for details). After the integration time of $10\tau$, all of the experimental results achieved a statistically steady state and agreed with the theoretical curve (compare the orange curve and the blue squares). As expected based on the analysis, the theoretical curve shows the scaling $\sigma_{S'} \sim L^{1/3}$ for large scales (approximately $L > 10^1$m) and the improper scaling $\sigma_{S'} \sim L^{2/3}$ for small scales (approximately $L < 10^{-1}$m). These results are contrary to the results

of DNSs and LESs (scaled-up DNSs) conducted by Thomas et al. (2020) (black circles in Figure 1), which show the proper scalings both for large and small scales ($\sigma_{S'} \sim L^{1/3}$ and $\sim L^1$, respectively).

    Note that Figure 1 also shows the results of the numerical integration of the eddy-hopping model reported by Thomas et al. (2020) (red triangles), and their results disagree with the results of the present study. A possible reason for this discrepancy might be that their results did not achieve a statistically steady state. For details, see Appendix C.

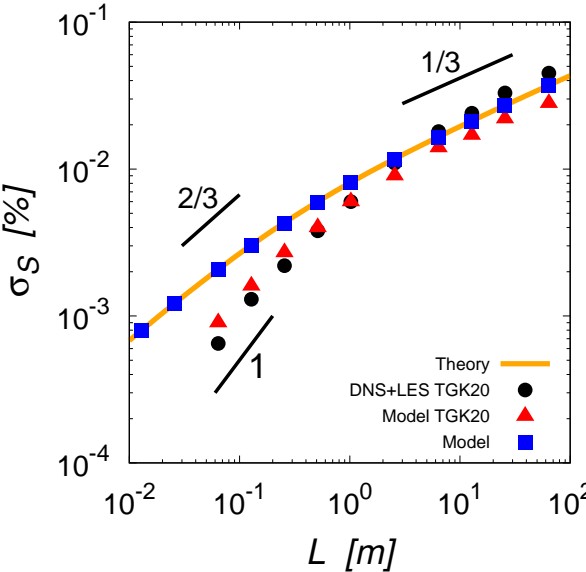

**Figure 1.** Standard deviation of the supersaturation fluctuation $\sigma_{S'}$ in the statistically steady state obtained from the analytical expression given by Eq. (13) (orange curve) and the results of our numerical integration of the eddy-hopping model (blue squares). The horizontal axis is the integral length $L$. The black circles indicate the reference data taken from direct numerical simulations and large-eddy simulations by Thomas et al. (2020). The red triangles indicate the results of the numerical integration of the eddy-hopping model reported by Thomas et al. (2020). The range of $L$ and $\sigma_{S'}$ for the panel is the same as in Figure 10 in Thomas et al. (2020). The three short black lines indicate slopes of 1, 2/3, and 1/3.

The eddy-hopping model given by Eqs. (1) and (2) shows the improper scaling for small scales because of the assumption made in the formulation of the model. Originally, Eq. (2) [corresponding to Eq. (8) in Grabowski and Abade (2017)] was formulated under the assumption of large scales (or $Da \gg 1$), since this assumption usually holds for typical situations in atmospheric clouds. Thus, it is reasonable that the eddy-hopping model does not reproduce the proper scaling for small scales. In the following section, we introduce some corrections to the original model given by Eqs. (1) and (2) so that the corrected

model can reproduce proper scalings both for large and small scales.


## 4 Corrections to the model

We introduce corrections to the eddy-hopping model so that the model can reproduce the reference data taken from DNSs and LESs in Figure 1. We first show the results. Equations (1) and (2) are, respectively, corrected as follows:

$$w'(t+\delta t) = w'(t)\mathrm{e}^{-\delta t/(c_1\tau)} + \sqrt{1 - \mathrm{e}^{-\frac{2\delta t}{(c_1\tau)}}}\,\sigma_{w'}\psi, \tag{18}$$

$$\frac{dS'}{dt} = a_1 w' - \frac{S'}{(c_2\tau_{relax})} - \frac{S'}{(c_1\tau)}. \tag{19}$$

There are two types of corrections. First, we added a term that is proportional to $-S'/\tau$ to Eq. (19). Physically, this term represents the damping effect on $S'$ due to turbulent mixing (eddy diffusivity). This type of term is commonly included in stochastic models used in cloud turbulence research (Sardina et al., 2015, 2018; Chandrakar et al., 2016; Siewert et al., 2017; Saito et al., 2019a). The time scale of the damping effect due to turbulent mixing is characterized by the integral time $\tau$, whereas that due to condensation/evaporation of cloud droplets is characterized by the phase relaxation time $\tau_{relax}$. The relative importance of these two effects is characterized by the Damköhler number ($Da = \tau/\tau_{relax}$), where the damping effect due to turbulent mixing is dominant for $Da \ll 1$ (corresponding to small scales). We also note that the term $-S'/\tau$ has been introduced to the equation for $S'$ in the eddy-hopping model by Abade et al. (2018) [see Eq. (15) of their study]. However, a subsequent study by Thomas et al. (2020) used a model without the term $-S'/\tau$ [see Eq. (14) of their study]. Although there might be some confusion, we again emphasize the importance of this term, which plays an essential role in reproducing the reference data, as described below.

The second correction to the model is that we changed two time scales, $\tau$ and $\tau_{relax}$, to $(c_1\tau)$ and $(c_2\tau_{relax})$, respectively. Here, $c_1$ and $c_2$ are constants and are used as tuning parameters. These types of parameters are not new. For example, a parameter corresponding to $c_1$ is commonly used in the Langevin stochastic equation in turbulence research (Sawford, 1991; Marcq and Naert, 1998). Formally, the inverse of $c_1$ is referred to as the drift coefficient, and the coefficients for the velocity and scalar equations should be distinguished. However, we treat these coefficients as the same parameter in Eqs. (18) and (19) for simplicity. On the other hand, the importance of a parameter corresponding to $c_2$ has been demonstrated in a recent study on turbulence modulation by particles (Saito et al., 2019b)

Applying the analytical procedure described in Section 3 to the corrected model given by Eqs. (18) and (19), we first obtain

$$\langle w'S'\rangle = a_1 \left(\frac{2}{c_1\tau} + \frac{1}{c_2\tau_{relax}}\right)^{-1}\sigma_{w'}^2$$

$$= c_1 a_1 [2 + (c_1/c_2)Da]^{-1}\tau\sigma_{w'}^2, \tag{20}$$

instead of Eq. (8). Next, instead of Eq. (11), we have

$$\sigma_{S'}^2 = \langle S'^2\rangle = a_1 \left(\frac{1}{c_1\tau} + \frac{1}{c_2\tau_{relax}}\right)^{-1}\langle w'S'\rangle$$

$$= c_1 a_1 [1 + (c_1/c_2)Da]^{-1}\tau\langle w'S'\rangle. \tag{21}$$

Finally, the analytical expression corresponding to Eq. (13) is

$$\sigma_{S'} = [1 + (c_1/c_2)Da]^{-1/2}[2 + (c_1/c_2)Da]^{-1/2}c_1 a_1 \tau\sigma_{w'}. \tag{22}$$





Asymptotic forms of $\sigma_{S'}$ in Eq. (22) for the large and small scale limits are, respectively, given as follows:

1. *Large scale limit*

    For $\tau \to \infty$ (or equivalently, $Da \to \infty$, $L \to \infty$), $\sigma_{S'}$ in Eq. (22) is approximated as

$$\begin{aligned} \sigma_{S'} &\approx c_2 a_1 Da^{-1} \tau \sigma_{w'} \\ &= c_2 a_1 \tau_{relax} \sigma_{w'}. \end{aligned} \tag{23}$$

   For the case of a constant dissipation rate of turbulent kinetic energy $\varepsilon$, we have

$$\sigma_{S'} \sim L^{1/3}. \tag{24}$$

2. *Small scale limit*

   For $\tau \to 0$ (or equivalently, $Da \to 0$, $L \to 0$), $\sigma_{S'}$ in Eq. (22) is approximated as

$$\sigma_{S'} \approx 2^{-1/2} c_1 a_1 \tau \sigma_{w'}, \tag{25}$$

   which indicates that $\tau_{relax}^{1/2}$ in Eq. (16) has been replaced by $\tau^{1/2}$ by introducing the term $-S'/\tau$ in Eq. (19). For the case of a constant dissipation rate of turbulent kinetic energy $\varepsilon$, we have

$$\sigma_{S'} \sim L. \tag{26}$$

Therefore, the corrected model reproduces asymptotic forms $\sigma_{S'} \sim a_1 \tau_{relax} \sigma_{w'}$ and $\sim a_1 \tau \sigma_{w'}$ for the large and small scale limits, respectively, which are both consistent with the result of the scaling argument by Lanotte et al. (2009).

Two parameters $c_1$ and $c_2$ are determined by comparing the theoretical curve given by Eq. (22) with the reference data taken from DNSs and LESs in Thomas et al. (2020). The best fit is given by $c_1 = 0.746$ and $c_2 = 1.28$. Figure 2 (orange curve) shows the theoretical curve given by Eq. (22) with these values of $c_1$ and $c_2$, which agrees well with the reference data (black circles).

Figure 2 (green diamonds) also shows the results of the numerical integration of the corrected model, which agree with the theoretical curve, as expected. Here, the numerical integration was conducted in the same manner as in the previous section (see Appendix A for details). These results confirm that the corrected eddy-hopping model given by Eqs. (18) and (19), with proper values of $c_1$ and $c_2$, can accurately reproduce the reference data.

## 5 Possibility of simplification of the model

Finally, we discuss the possibility of simplification of the eddy-hopping model. Here, our discussion is based on the corrected model given by Eqs. (18) and (19), but the same argument also applies to the original model given by Eqs. (1) and (2).

The eddy-hopping model consists of two evolution equations for the supersaturation and vertical velocity fluctuations, $S'$ and $w'$ respectively, and these two variables fluctuate randomly according to the Ornstein-Uhlenbeck process. However, if we



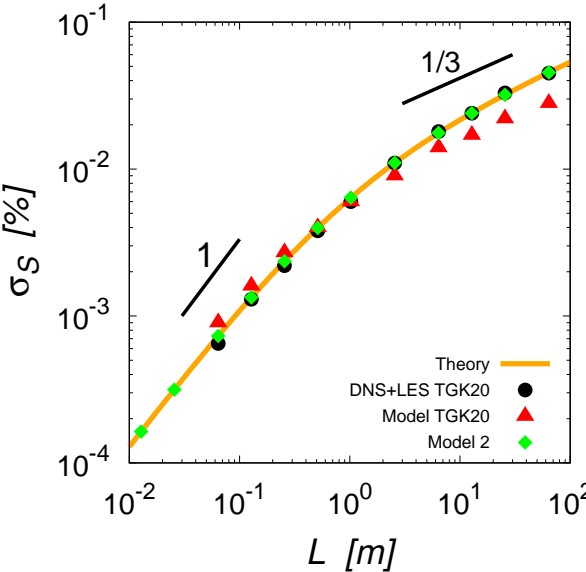

**Figure 2.** Standard deviation of the supersaturation fluctuation $\sigma_{S'}$ in the statistically steady state obtained from the analytical expression given by Eq. (22) for the corrected model (orange curve) and the results of our numerical integration using the corrected model given by Eqs. (18) and (19) (green diamonds). Here, the two tuning parameters are set to $c_1 = 0.746$ and $c_2 = 1.28$. The two short black lines indicate slopes of 1 and $1/3$. The black circles and the axes of the panel are the same as in Figure 1.

have $S'$ that fluctuates with a proper amplitude and auto-correlation function, then we do not need the evolution equation for

$w'$, because only $S'$ is used in the growth equation of the droplet size. As described in Section 4, we obtained an analytical expression for $\sigma_{S'}$, i.e., the standard deviation of the supersaturation fluctuation in the statistically steady state given by Eq. (22). On the other hand, the auto-correlation function for $S'$ in a statistically steady state can also be obtained analytically. The derivation is described in Appendix D. The result is given in Eq. (D14) and is as follows:

$$A(t) = \frac{\langle S'(t+t_0)S'(t_0)\rangle}{\langle S'(t_0)S'(t_0)\rangle} \tag{27}$$

$$= \left(\frac{\tau_1}{\tau_1 - \tau_2}\right)e^{-t/\tau_1} - \left(\frac{\tau_2}{\tau_1 - \tau_2}\right)e^{-t/\tau_2}, \tag{28}$$

where $\tau_1$ and $\tau_2$ are, respectively, defined as

$$\tau_1 = c_1\tau, \quad \text{and} \quad \tau_2 = \left(\frac{1}{c_1\tau} + \frac{1}{c_2\tau_{relax}}\right)^{-1} = c_1\tau\left(1 + \frac{c_1}{c_2}Da\right)^{-1}. \tag{29}$$

We can also obtain the auto-correlation time for $S'$ by time integration of $A(t)$ [see Eq. (D16) in Appendix D], which is given as

$$\tau_0 = \tau_1 + \tau_2, \tag{30}$$





The auto-correlation function $A(t)$ in Eq. (28) and the auto-correlation time $\tau_0$ in Eq. (30), with $\tau_1$ and $\tau_2$ defined by (29), have important asymptotic forms in two limits. First, for the large scale limit ($Da \to \infty$), the asymptotic forms of $A(t)$ and $\tau_0$ are given by

$$\lim_{Da \to \infty} A(t) = e^{-t/(c_1\tau)} \qquad \text{and} \qquad \lim_{Da \to \infty} \tau_0 = c_1\tau, \tag{31}$$

respectively. Second, for the small scale limit ($Da \to 0$), the asymptotic forms of $A(t)$ and $\tau_0$ are given by

$$\lim_{Da \to 0} A(t) = \left[1 + \frac{t}{(c_1\tau)}\right] e^{-t/(c_1\tau)} \qquad \text{and} \qquad \lim_{Da \to 0} \tau_0 = 2c_1\tau, \tag{32}$$

respectively.

Based on analytical expressions for the fluctuation amplitude and the auto-correlation function for $S'$ [Eqs. (22) and (30), respectively], a simplified version of the eddy-hopping model is defined as follows:

$$S'(t + \delta t) = S'(t)e^{-\delta t/\tau_0} + \sqrt{1 - e^{-\frac{2\delta t}{\tau_0}}}\, \sigma_{S'}\psi, \tag{33}$$

where $\sigma_{S'}$ and $\tau_0$ are given by Eqs. (22) and (30), respectively. Note that the simplified model given by Eq. (33) is a single-equation model, as compared to the two-equation model given by Eqs. (18) and (19) before the simplification. The auto-correlation function for $S'$ in the simplified model given by Eq. (33) is given by

$$B(t) \quad = \quad \frac{\langle S'(t + t_0)S'(t_0)\rangle}{\langle S'(t_0)S'(t_0)\rangle} \tag{34}$$

$$\quad = \quad e^{-t/\tau_0}, \tag{35}$$

which has the following two asymptotic forms. First, for the large scale limit ($Da \to \infty$),

$$\lim_{Da \to \infty} B(t) = e^{-t/(c_1\tau)}, \tag{36}$$

which agrees with the corresponding asymptotic form given by Eq. (31) for the corrected model. Second, for the small scale limit ($Da \to 0$),

$$\lim_{Da \to 0} B(t) = e^{-t/(2c_1\tau)}, \tag{37}$$

which disagrees with the corresponding asymptotic form given by Eq. (32) for the corrected model.

Figures 3(a) through 3(e) compare the auto-correlation function for the simplified model [$B(t)$ in Eq. (35): blue dashed curve] and that for the corrected model [$A(t)$ in Eq. (28): red solid curve] for five cases ranging from $Da \ll 1$ to $Da \gg 1$. Note that the time $t$ is normalized by the auto-correlation time $\tau_0$ for each case. Although $B(t)$ and $A(t)$ share the same auto-

correlation time, $B(t)$ deviates from $A(t)$ for cases with $Da$ of order unity or smaller, as shown in Figures 3(a) through 3(c). On the other hand, for $Da \gg 1$, $B(t)$ agrees with $A(t)$ very well, as shown in Figures 3(d) and 3(e).

The advantages of the simplified version of the eddy-hopping model given by Eq. (33) are summarized as follows.

1. *Reduction of computational cost.* Compared to the corrected model, which consists of two equations [Eqs. (18) and (19) for $w'$ and $S'$, respectively], the simplified model has a single equation [Eq. (33) for $S'$]. If we consider the implemen-

tation of the eddy-hopping model to the LES Lagrangian cloud model based on the super-droplet method (Shima et al.,



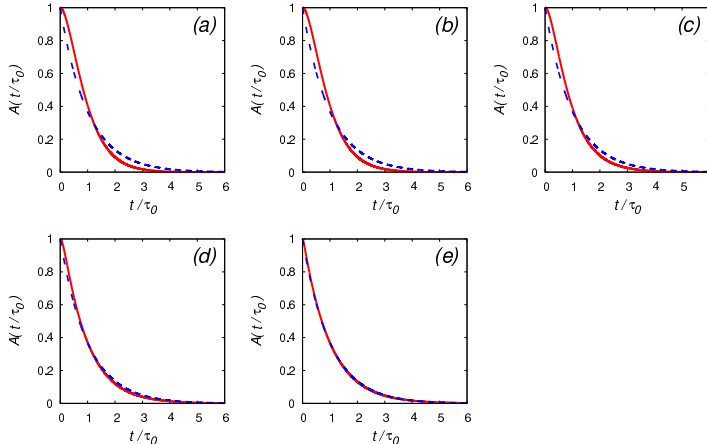

**Figure 3.** Auto-correlation functions in the statistically steady state for the simplified model [$B(t)$ in Eq. (35): blue dashed curve] and the corrected model [$A(t)$ in Eq. (28): red solid curve]. The parameters for each panel are as follows: (a) $L = 10^{-2}$ m, $\tau = 0.447$ s, $\tau_0 = 0.644$ s, $Da = 0.127$, (b) $L = 10^{-1}$ m, $\tau = 2.08$ s, $\tau_0 = 2.70$ s, $Da = 0.591$, (c) $L = 10^0$ m, $\tau = 9.63$ s, $\tau_0 = 9.95$ s, $Da = 2.74$, (d) $L = 10^1$ m, $\tau = 44.7$ s, $\tau_0 = 37.3$ s, $Da = 12.7$, and (e) $L = 10^2$ m, $\tau = 208$ s, $\tau_0 = 159$ s, $Da = 59.1$. The phase relaxation time is fixed to $\tau_{relax} = 3.513$ s. The horizontal axis is the time $t$ normalized by the auto-correlation time $\tau_0$ for each case. The parameter $\tau$ is determined from the integral length $L$ based on the setting for the numerical experiment described in Appendices A and B.

2009), then this difference means that the number of the additional attribute (particle variable) is smaller for the simplified model, which contributes to a reduction in computational cost.

2. *Desirable convergence property.* The auto-correlation function for the simplified model [$B(t)$ in Eq. (35)] converges to that for the corrected model in the large-scale limit ($Da \to \infty$), as shown in Eq. (36). As confirmed in Figures 3(d) and 3(e), the two auto-correlation functions are almost identical for an integral length $L$ greater than 10 m (or $Da \geq 10$). In the implementation of the eddy-hopping model to the LES Lagrangian cloud model, the integral length $L$ is supposed to roughly correspond to the grid size, which is often greater than several meters to several tens of meters. Therefore, the assumption of large scales (or $Da \gg 1$) usually holds, in which case the statistical properties of the simplified model are expected to be almost unchanged after the simplification.

# 6 Summary and conclusions

The purpose of the present paper was to obtain various statistical properties of the eddy-hopping model, a novel cloud microphysical model, which accounts for the effect of the supersaturation fluctuation at unresolved scales on the growth of cloud droplets and on spectral broadening. Based on derived statistical properties, we first showed in Section 3 that the model fails to reproduce a proper scaling for smaller Damköhler numbers (corresponding to small scales), resulting in a deviation of the model prediction from the reference data taken from DNSs and LESs, as shown in Figure 1. In Section 4, we introduced two





corrections to the model so that the corrected can accurately reproduce the reference data with the proper scaling. The first correction is to add the term representing the effect of turbulent mixing (eddy diffusivity) on the supersaturation fluctuation, and the second is to multiply time scales $\tau$ (integral time) and $\tau_{relax}$ (phase relaxation time) by tuning parameters $c_1$ and $c_2$, respectively. In Section 5, we discussed the possibility of simplification of the model. The simplified model consists of a single stochastic equation for the supersaturation fluctuation, as in Eq. (33), with amplitude and time parameters given by the corresponding analytical expressions for the model before the simplification. We showed that, for larger Damköhler numbers (corresponding to large scales), the auto-correlation function of the supersaturation fluctuation for the simplified model converges to that for the model before the simplification. Since the assumption of large scales usually holds in the typical parameter range for the model implementation in the LES Lagrangian cloud model, the simplified model may contribute to a reduction in computational cost while keeping the statistical properties almost unchanged after the simplification.

In a future study, the actual performance of the simplified model should be validated through numerical simulations of cloud models, such as those by Grabowski and Abade (2017); Abade et al. (2018).

**Appendix A: Numerical integration of the eddy-hopping model**

The results of the numerical integration of the eddy-hopping model [Eqs. (1) and (2)] and the corrected eddy-hopping model [Eqs. (18) and (19)] are shown in Figures 1 (blue squares) and 2 (green diamonds), respectively. For these experiments, we used the same setting as that in Section 5 in Thomas et al. (2020), except that the integration time was increased from $6\tau$ to $10\tau$. We set $a_1 = 4.753 \times 10^{-4}$ m$^{-1}$, $\varepsilon = 10$ cm$^2$ s$^{-3}$, and $\tau_{relax} = 3.513$ s, and the integral time $\tau$ as

$$\tau = \frac{1}{(2\pi)^{1/3}}\left(\frac{L}{\sigma_{w'}}\right). \tag{A1}$$

As described in Appendix B, for the case of a constant dissipation rate of turbulent kinetic energy $\varepsilon$, $\sigma_{w'}$ is given as a function of $L$. We time integrated the governing equations of the model using 12 values of $L$: $L = 0.0128, 0.0256, 0.064, 0.128, 0.256,$ $0.512, 1.024, 2.56, 6.4, 12.8, 25.6,$ and $64.0$ m. The time step $\delta t$ is set as 1/1,000 of $\tau$, and the integration time is $10\tau$. In addition, the numerical scheme is the forward Euler method. The initial condition is such that $w'(0) = \sigma_{w'}\psi$ and $S'(0) = 0$. Each result in Figures 1 (blue squares) and 2 (green diamonds) is obtained by averaging the results for 1,000 ensembles with different seeds of random numbers.

**Appendix B: Scalings for the case of a constant dissipation rate of turbulent kinetic energy**

We consider classical homogeneous isotropic turbulence, in which energy is mainly injected into the system at large scales, cascaded to smaller scales by nonlinear interaction, and finally dissipated by the molecular viscosity in the smallest scales. In a statistically steady state, the dissipation rate of turbulent kinetic energy is defined as $\varepsilon$. If $\varepsilon$ is fixed and the integral scale $L$ is changed, then the kinetic energy $E$ scales as follows (Thomas et al., 2020):

$$E \sim (L\varepsilon)^{2/3}. \tag{B1}$$



The black circles in Figure B1 show the relation between $L$ and $E$ in the reference data taken from DNSs and LESs by Thomas et al. (2020) (Table 2 of their study). In their simulation, the dissipation rate was fixed to $\varepsilon = 10 \text{ cm}^2 \text{s}^{-3}$. The orange curve in Figure B1 indicates the function $E = \alpha \varepsilon^{2/3} L^{2/3}$, where $\alpha$ is the fitting parameter. The best fit is given by $\alpha = 0.475$. The root-mean-square turbulent velocity is calculated as a function of $L$ by $u_{rms} = \sigma_{w'} = \sqrt{(2E/3)}$, and $\sigma_{w'}$ is used as the

parameter in the eddy-hopping model. Note that Thomas et al. (2020) used the same type of large-scale forcing as that used by Kumar et al. (2012), where the integral length $L$ is set to be equal to the box length $L_{box}$.

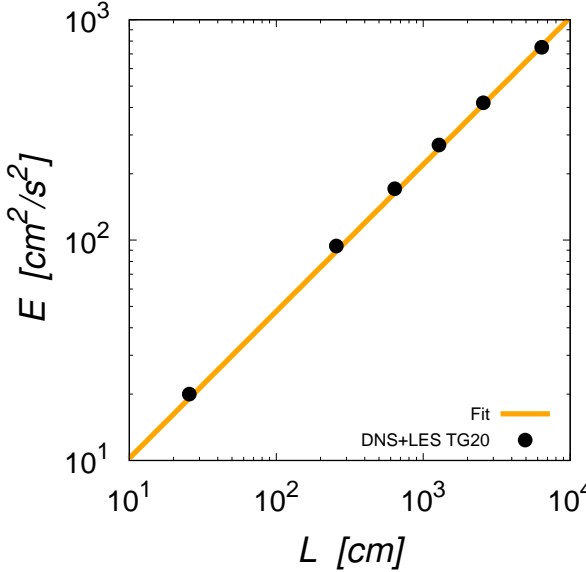

**Figure B1.** Relationship between the integral scale $L$ and the turbulent kinetic energy $E$. The black circles are taken from the reference data in Thomas et al. (2020). The orange curve indicates the fitting function $E = \alpha \varepsilon^{2/3} L^{2/3}$ with $\alpha = 0.475$.

**Appendix C: Achievement of a statistically steady state**

We confirm that all of the results of the numerical integration of the eddy-hopping model in the present study achieved statistically steady states. For this purpose, we first derive the analytical expression for the time evolutions of the variance and

covariance of the variables in the model and then compare these analytical expressions with the results of the numerical integration.

The governing equations given by Eqs. (3) and (2) can be rewritten in generalized forms as

$$\frac{dw'}{dt} = -\frac{1}{\tau_1} w'(t) + F_{w'}(t), \tag{C1}$$

$$\frac{dS'}{dt} = a_1 w'(t) - \frac{S'(t)}{\tau_2}, \tag{C2}$$





where $\tau_1$ and $\tau_2$ are the relaxation times for $w'$ and $S'$, respectively, and the forcing term $F_{w'}(t)$ satisfies Eq. (4). Evolution equations for the variance and covariance of the variables are derived as follows:

$$\frac{dV_{w'}(t)}{dt} = -\frac{2}{\tau_1}V_{w'}(t) + \left(\frac{2\sigma_{w'}^2}{\tau_1}\right), \tag{C3}$$

$$\frac{dC(t)}{dt} = a_1 V_{w'}(t) - \left(\frac{1}{\tau_1} + \frac{1}{\tau_2}\right)C(t), \tag{C4}$$

$$\frac{dV_{S'}(t)}{dt} = -\frac{2}{\tau_2}V_{S'}(t) + 2a_1 C(t), \tag{C5}$$

where $V_{w'}(t)$, $C(t)$, and $V_{S'}(t)$ are, respectively, defined as

$$V_{w'}(t) = \langle w'(t)w'(t)\rangle, \tag{C6}$$

$$C(t) = \langle w'(t)S'(t)\rangle, \tag{C7}$$

$$V_{S'}(t) = \langle S'(t)S'(t)\rangle. \tag{C8}$$

For the numerical integration of the eddy-hopping model by Thomas et al. (2020), $\tau_1 = \tau$ and $\tau_2 = \tau_{relax}$. Since the initial con-
ditions for $w'(t)$ and $S'(t)$ are set to $w'(0) = \sigma_{w'}\psi$ and $S'(0) = 0$ in Thomas et al. (2020), the corresponding initial conditions for the variance and covariance are given by

$$V_{w'}(0) = \sigma_{w'}^2, \tag{C9}$$

$$C(0) = 0, \tag{C10}$$

$$V_{S'}(0) = 0. \tag{C11}$$

Solving Eqs. (C3) through (C5) with the initial conditions given by Eqs. (C9) through (C11), we obtain

$$V_{w'}(t) = \sigma_{w'}^2, \tag{C12}$$

$$C(t) = a_1 \sigma_{w'}^2 \tau_3 \left(1 - e^{-t/\tau_3}\right), \tag{C13}$$

$$V_{S'}(t) = a_1^2 \sigma_{w'}^2 \tau_3 \tau_2 \left(1 - e^{-2t/\tau_2}\right) + 2a_1^2 \sigma_{w'}^2 \tau_3 \tau_4 \left(e^{-t/\tau_3} - e^{-2t/\tau_2}\right), \tag{C14}$$

where $\tau_3$ and $\tau_4$ are, respectively, defined as

$$\tau_3 = \frac{\tau_1 \tau_2}{\tau_1 + \tau_2}, \qquad \text{and} \qquad \tau_4 = \frac{\tau_1 \tau_2}{\tau_2 - \tau_1}. \tag{C15}$$

  Figure C1 compares the analytical expression given by Eq. (C14) (cyan circles) with the results of the numerical integration of the eddy-hopping model given by Eqs. (1) and (2) (black crosses). The setting for the numerical experiment is the same as that used in Figure 1, except that the integration time is $0.6\tau$ in Figure C1(a) and $6\tau$ in Figure C1(b). The results of the numerical integration (black crosses) agree well with the analytical expression (cyan circles), and both approach the theoretical
curve for the statistically steady state (orange curve in each panel) as the integration time increases. Figure C1(a) also indicates that the results of the numerical integration of the eddy-hopping model by Thomas et al. (2020) (red triangles) are fairly close to our results at $0.6\tau$. Thus, it might be possible that the integration time of their numerical experiment was not long enough to achieve a statistically steady state.

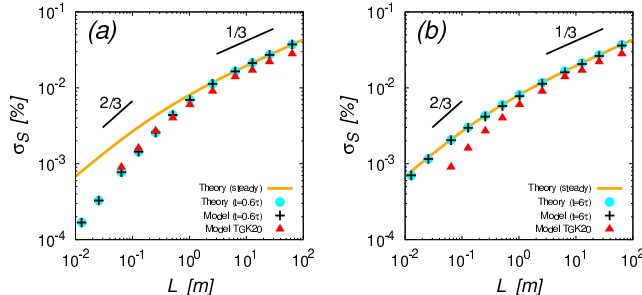

**Figure C1.** Standard deviation of the supersaturation fluctuation $\sigma_{S'}$ at times $t = 0.6\tau$ [panel (a)] and $t = 6\tau$ [panel (b)] obtained from the analytical expression given by Eq. (C14) (cyan circles) and the results of the numerical integration of the eddy-hopping model given by Eqs. (1) and (2) (black crosses) The orange curve, red triangles, and axes of the panel are the same as in Figure 1. The two short black lines indicate slopes of $2/3$ and $1/3$. The setting for the numerical integration is the same as that used in Section 3, except that the integration times are $0.6\tau$ and $6\tau$ in (a) and (b), respectively.

**Appendix D: Derivation of auto-correlation function**

We derive the analytical expression for the auto-correlation function of the supersaturation fluctuation $S'(t)$ in the eddy-hopping model. As in Appendix C, we start from the generalized form of the eddy-hopping model as follows:

$$\frac{dw'}{dt} = -\frac{1}{\tau_1}w'(t) + F_{w'}(t), \tag{D1}$$

$$\frac{dS'}{dt} = a_1 w'(t) - \frac{S'(t)}{\tau_2}, \tag{D2}$$

where $\tau_1$ and $\tau_2$ are the relaxation times for $w'$ and $S'$, respectively, and the forcing term $F_{w'}(t)$ satisfies Eq. (4). We consider

that the system is in a statistically steady state.

First, multiplying Eq. (D2) by $\mathrm{e}^{t/\tau_2}$ and applying the product rule of differentiation, we obtain

$$\frac{d}{dt}\left(S'(t)\mathrm{e}^{t/\tau_2}\right) = a_1 w'(t)\mathrm{e}^{t/\tau_2}. \tag{D3}$$

Integrating Eq. (D3) from $t = 0$ to $t$, we obtain

$$S'(t) = S'(0)\mathrm{e}^{-t/\tau_2} + \int_0^t a_1 w'(\xi)\mathrm{e}^{(\xi-t)/\tau_2}d\xi. \tag{D4}$$

(Note that we chose the integration range $[0,t]$ for simplicity of notation. Since we consider a statistically steady state, the following discussion is unchanged if the integration range is $[t_0, t_0 + t]$.) Applying a similar procedure as above to Eq. (D1) with the integration range $t : 0 \rightarrow \xi$, we obtain

$$w'(\xi) = w'(0)\mathrm{e}^{-\xi/\tau_1} + \int_0^\xi F_{w'}(\zeta)\mathrm{e}^{(\zeta-\xi)/\tau_1}d\zeta. \tag{D5}$$





Substituting Eq. (D5) into Eq. (D4) and calculating some of the integrations, we obtain

$$
\begin{aligned}
\quad S'(t) &= S'(0)\mathrm{e}^{-t/\tau_2} + \int_0^t a_1 \left( w'(0)\mathrm{e}^{-\xi/\tau_1} + \int_0^\xi F_{w'}(\zeta)\mathrm{e}^{(\zeta-\xi)/\tau_1}d\zeta \right) \mathrm{e}^{(\xi-t)/\tau_2}d\xi \tag{D6}\\[2mm]
&= S'(0)\mathrm{e}^{-t/\tau_2} + a_1 w'(0)\mathrm{e}^{-t/\tau_2}\int_0^t \mathrm{e}^{(\tau_2^{-1}-\tau_1^{-1})\xi}d\xi + a_1\mathrm{e}^{-t/\tau_2}\int_0^t\int_0^\xi F_{w'}(\zeta)\mathrm{e}^{\zeta/\tau_1}\mathrm{e}^{(\tau_2^{-1}-\tau_1^{-1})\xi}d\zeta d\xi \tag{D7}\\[2mm]
&= S'(0)\mathrm{e}^{-t/\tau_2} + a_1 w'(0)\left(\tau_2^{-1}-\tau_1^{-1}\right)^{-1}\left(\mathrm{e}^{-t/\tau_1}-\mathrm{e}^{-t/\tau_2}\right) + a_1\mathrm{e}^{-t/\tau_2}\int_0^t\int_0^\xi F_{w'}(\zeta)\mathrm{e}^{\zeta/\tau_1}\mathrm{e}^{(\tau_2^{-1}-\tau_1^{-1})\xi}d\zeta d\xi. \tag{D8}
\end{aligned}
$$

Multiplying Eq. (D8) by $S'(0)$ and taking an ensemble average, we obtain

$$
\langle S'(t)S'(0)\rangle = \langle S'(0)S'(0)\rangle\mathrm{e}^{-t/\tau_2} + a_1\langle w'(0)S'(0)\rangle\left(\tau_2^{-1}-\tau_1^{-1}\right)^{-1}\left(\mathrm{e}^{-t/\tau_1}-\mathrm{e}^{-t/\tau_2}\right), \tag{D9}
$$

because of the statistical independence ($\langle F_{w'}(\zeta)S'(0)\rangle = 0$). Next, as in the derivation of Eq. (11), we multiply Eq. (D2) by $S'$ and consider the statistically steady state. We obtain

$$
\langle S'(0)S'(0)\rangle = a_1\tau_2\langle w'(0)S'(0)\rangle. \tag{D10}
$$

Substituting Eq. (D10) into Eq. (D9), we have

$$
\langle S'(t)S'(0)\rangle = \langle S'(0)S'(0)\rangle\mathrm{e}^{-t/\tau_2} + \langle S'(0)S'(0)\rangle\tau_2^{-1}\left(\tau_2^{-1}-\tau_1^{-1}\right)^{-1}\left(\mathrm{e}^{-t/\tau_1}-\mathrm{e}^{-t/\tau_2}\right). \tag{D11}
$$

Therefore, the auto-correlation function of the supersaturation fluctuation $S'(t)$ for the eddy-hopping model given by Eqs. (D1) and (D2) in the statistically steady state is written as follows:

$$
\begin{aligned}
A(t) &= \frac{\langle S'(t)S'(0)\rangle}{\langle S'(0)S'(0)\rangle} \tag{D12}\\[2mm]
&= \mathrm{e}^{-t/\tau_2} + \frac{\tau_1}{\tau_1-\tau_2}\left(\mathrm{e}^{-t/\tau_1}-\mathrm{e}^{-t/\tau_2}\right) \tag{D13}\\[2mm]
&= \left(\frac{\tau_1}{\tau_1-\tau_2}\right)\mathrm{e}^{-t/\tau_1} - \left(\frac{\tau_2}{\tau_1-\tau_2}\right)\mathrm{e}^{-t/\tau_2} \tag{D14}
\end{aligned}
$$

The auto-correlation time $\tau_0$ is obtained by time-integrating $A(t)$ as

$$
\begin{aligned}
\tau_0 &= \int_0^\infty A(t)dt = \frac{\tau_1^2-\tau_2^2}{\tau_1-\tau_2} \tag{D15}\\[2mm]
&= \tau_1 + \tau_2. \tag{D16}
\end{aligned}
$$

For the original version of the eddy-hopping model given by Eqs. (1) and (2), we have

$$
\tau_1 = \tau, \quad \text{and} \quad \tau_2 = \tau_{relax}. \tag{D17}
$$

For the corrected version given by Eqs. (18) and (19), we have

$$
\tau_1 = c_1\tau, \quad \text{and} \quad \tau_2 = \left(\frac{1}{c_1\tau} + \frac{1}{c_2\tau_{relax}}\right)^{-1}. \tag{D18}
$$





*Author contributions.* IS conducted the numerical simulations and data analysis. IS and TW performed the theoretical analyses in Sections 3 and 4. IS and TG performed the theoretical analysis in Section 5. All three of the authors were involved in preparing the manuscript.

*Competing interests.* The authors declare that they have no conflicts of interest.

*Acknowledgements.* We are grateful to Kei Nakajima for his technical support. The present study was supported by MEXT KAKENHI Grants No. 20H00225 and 20H02066, by JSPS KAKENHI Grant No. 18K03925, by the Naito Foundation, by the HPCI System Research Project (Project ID: hp200072), by the NIFS Collaboration Research Program (NIFS20KNSS143), by the Japan High Performance Computing and Networking plus Large-scale Data Analyzing and Information Systems (jh200006), and by High Performance Computing (HPC 2020) at Nagoya University.





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
