# Peer review of "Statistical properties of a stochastic model of eddy hopping"

_Atmospheric Chemistry and Physics, 2021_

## Author Comment (AC1)

**Reply to Referee 3**

We are deeply grateful for the referee's comments on our paper. Following your comments, we revised the manuscript. Our responses to your comments are as follows. Lines are those in the revised version. For convenience, we attach a supplemental material which is the same as the revised manuscript except that the changed parts are written in red color.

1. *The authors suggest that the current "eddy hopping model" has limitations in representing the correct scaling at small scales. A correction to the model is introduced (an additional drift term representing the turbulent mixing). A further modification is introduced by multiplying the timescales with constant coefficients that adjust the magnitude of the supersaturation variance. One of the corrections was already included in earlier papers (additional drift term from turbulent mixing), although the evaluation of the supersaturation variance scaling was not presented before.*

   Thanks for the comment. This point ["*One of the corrections was already included in earlier papers (additional drift term from turbulent mixing)*"] was also mentioned by the other referee. Additional drift term representing for turbulent mixing was first introduced into the eddy-hopping model by Abade et al. (2018), and therefore our contribution should be stated as validation of the model, rather than "correction" to the model. Following this comment, we made revisions as described below.

   In the revised manuscript, we first made the following revisions to correctly describe the contribution by Abade et al. (2018):

   - Line 21–23: A sentence "*Abade et al. (2018) extended the model ... due to turbulent mixing.*" has been inserted.
   - Line 125, section 4: The section title has been changed.
   - Line 126: A sentence "*We next consider ... as follows:*" has been inserted.
   - Line 131–132: A sentence "*The important change ... in Eq. (19).*" has been inserted.

   We made the following revision to distinguish two versions of the model:

   - Line 23–24: A sentence "*For clarity, we ... the second version.*" has been inserted.

   Accordingly, we removed the word and the phrase such as "corrected", "corrected model", and so on, and we refer to the eddy-hopping model developed by Grabowski & Abade (2017) as the "original version", and the extended model by Abade et al. (2018) as the "second version" in the revised manuscript. Also, to correctly describe our contribution, we made the following revisions:

   - Line 4–6: A sentence "*Two versions ... simulations (LESs).*" has been inserted.
   - Line 31–36: Three sentences "*These statistical properties ... leads to improvement.*" have been inserted.
   - Line 254–256: Two sentences "*we showed that ... the reference data.*" have been inserted.

2. *The new simplified model for the super-droplet method is good. But it might not provide much computational benefit if the subgrid-scale transport of super-droplets is also needed.*

Thanks for the comment. This was also pointed out by the other referee. Indeed, the simplified model does not necessarily lead to a reduction in computational cost when not only $S'$ but also $w'$ is used for the subgrid-scale parameterization in LES.

In the revised manuscript, we removed the phrase such as "reduction in computational cost" and instead used the phrase such as "reduces the number of model variables". We also removed the discussion on computational cost of the simplified model in section 5 and only discusses the convergence property of the model. Revisions are as follows:

- Line 7–8, Line 36: A phrase "*which may contribute to a reduction in computational cost*" has been replaced by "*which reduces the number of model variables*".
- Line 230: The item discussing possible reduction of computational cost ("*1. Reduction of computational cost. ... reduction in computational cost.*") has been removed.
- Line 260–262: The sentence "*Since the assumption ... after the simplification.*" has been replaced by "*This convergence property ... Lagrangian cloud model.*".

3. *L120-125 and Eq. 19: I agree the drift term due to turbulent mixing is necessary for correctly representing the supersaturation fluctuations. In fact, it was included in some of the past studies. However, a corresponding complementary diffusion term (the Wiener increment term) representing small-scale fluctuations/mixing would also be required in the corrected model.*

Thanks for the suggestion. Yes. It would be possible to further extend the second version of the eddy-hopping model [by Abade et al. (2018, JAS)] by additionally introducing the Wiener process term which represents small-scale fluctuations/mixing. Such terms are actually included in the Langevin model of the supersaturation fluctuation considered in the previous studies: for example, Eq. (42) in Paoli & Shariff (2009, JAS) and Eq. (7) in Sardina et al. (2015, PRL). We made the following revision to note this point:

- Line 139–142: A paragraph "*Note that it ... for future work.*" has been inserted.

In the present study, however, we focus on statistical properties of the the second version with Eqs. (18) and (19) and leave this extension for future work.

4. *L132-136: Are the drift coefficients introduced just to scale the magnitude of the supersaturation fluctuations to a correct value, or are there any other physical reasons? More explanation would be helpful for the readers.*

There are no other physical reasons. Two parameters $c_1$ and $c_2$ are introduced just to scale the magnitude of $S'$ to that of the reference data. To emphasize this point, we made the following revision:

- Line 181: A sentence "*Here, we do not ... as tuning parameters.*" has been inserted.

5. *A figure showing sample supersaturation trajectories from all three models (original, corrected, and simplified) could be informative (probably in the appendix section).*

Thanks for the suggestion. We made the following revisions to show sample supersaturation trajectories from three models:

- Page 12: Figure 5 has been inserted.

- Line 237–246: Two paragraphs "Figure 5 compares ... almost identical results." have been added.

6. *It would also be good to discuss the limitation of the current approach in representing the supersaturation fluctuation generation from scalar mixing (e.g., during the turbulent entrainment-mixing).*

Thanks for the comment. As the referee pointed out, the turbulent entrainment-mixing is another important mechanism for the supersaturation fluctuation generation other than the stochastic condensation, and the effects of the turbulent entrainment-mixing are not included in the eddy-hopping model considered in the present study. We made the following revision to note this point:

- Line 26–29: A paragraph "*It should be noted ... entraining parcel model.*" has been inserted.

We also referred to the paper Abade et al. (2018, JAS), which investigated the effects of the turbulent entrainment-mixing and entrained CCN activation by using the entraining parcel model.

We again appreciate the referee's valuable comments which are very constructive to make the paper clearer and better.

---

## Author Comment (AC2)

**Reply to Referee 2**

We are deeply grateful for the referee's comments on our paper. Following your comments, we revised the manuscript. Our responses to your comments are as follows. Lines are those in the revised version. For convenience, we attach a supplemental material which is the same as the revised manuscript except that the changed parts are written in red color.

1. ... the Authors of the submitted manuscript claim to have corrected the eddy hopping model of GA17 by including a relaxation term that have been already included and studied by AGP18... the introduction of tuning parameters to fit the model to reference data cannot be regarded as a "correction" to the model (particularly when those fitting parameters turn out to be O(1)), unless theoretical expressions for the parameters and a rationale are provided. ...the Authors should state differently their contribution to the field.

Thanks for the comment. We agree. Our main contribution is that we validated the eddy-hopping model [the original version by Grabowski & Abade (2017) and the extended version by Abade et al. (2018)], and not that we "corrected" the model.

In the revised manuscript, we first made the following revisions to correctly describe the contribution by Abade et al. (2018):

- Line 21–23: A sentence "Abade et al. (2018) extended the model ... due to turbulent mixing." has been inserted.
- Line 125, section 4: The section title has been changed.
- Line 126: A sentence "We next consider ... as follows:" has been inserted.
- Line 131–132: A sentence "The important change ... in Eq. (19)." has been inserted.

We made the following revision to distinguish two versions of the model:

• Line 23-24: A sentence "For clarity, we ... the second version." has been inserted.

Accordingly, we removed the word and the phrase such as "corrected", "corrected model", and so on, and we refer to the eddy-hopping model developed by Grabowski & Abade (2017) as the "original version", and the extended model by Abade et al. (2018) as the "second version" in the revised manuscript. Also, to correctly describe our contribution, we made the following revisions:

- Line 4–6: A sentence "Two versions ... simulations (LESs)." has been inserted.
- Line 31–36: Three sentences "These statistical properties ... leads to improvement." have been inserted.
- Line 254–256: Two sentences "we showed that ... the reference data." have been inserted.
- 2. Finally, the Authors suggest a simplification of the model that eliminates the vertical velocity fluctuations w' from the list of stochastic variables attached to computational particles (superdroplets) in LES Lagrangian cloud models. The discussion and procedure of model simplification is instructive and gives valuable insights into the model. However, this simplification does not necessarily imply in reduction of computational cost. This is because the velocity fluctuations of superdroplets are usually necessary to resolve the subgrid-scale

transport of superdroplets. Also, the simplified model accurately reproduces time correlations in the Lagrangian supersaturation in the regime of large Damkholer numbers. This is exactly the regime of applicability of the GA17 model.

Thanks for the comment. You are right. The simplified model does not necessarily lead to a reduction in computational cost when not only S' but also w' is used for the subgrid-scale parameterization in LES.

In the revised manuscript, we removed the phrase such as "reduction in computational cost" and instead used the phrase such as "reduces the number of model variables". We also removed the discussion on computational cost of the simplified model in section 5 and only discussed the convergence property of the model. Revisions are as follows:

- Line 7–8, Line 36: A phrase "which may contribute to a reduction in computational cost" has been replaced by "which reduces the number of model variables".
- Line 230: The item discussing possible reduction of computational cost ("1. Reduction of computational cost. ... reduction in computational cost.") has been removed.
- Line 260–262: The sentence "Since the assumption ... after the simplification." has been replaced by "This convergence property ... Lagrangian cloud model.".

We again appreciate the referee's valuable comments which are very instructive to make the paper clearer and better.

---

## Author Response (AR2)

**Reply to Referee 3**

We are deeply grateful for the referee's comments on our paper. Following your comments, we revised the manuscript. Our responses to your comments are as follows. Lines are those in the revised version.

1. *The authors have addressed most of my comments appropriately. I have an additional suggestion (not critical for the acceptance of the paper). The authors can also refer to the following recently appeared JAS article: "Impact of entrainment-mixing and turbulent fluctuations on droplet size distributions in a cumulus cloud: An investigation using Lagrangian microphysics with a sub-grid- scale model" by Chandrakar et al. JAS (2021). They used a similar subgrid model as Abade et al. (2018) but with a Wiener diffusion term to make the stochastic differential equation consistent with the drift-diffusion model and produce correct fluctuation magnitude and variability.*

   Thank you very much for suggesting the reference (Chandrakar et al. JAS 2021). This paper was published just recently and we did not know it. We made the following revisions and cited the paper.

   - Line 139: The phrase "*it would be possible*" has been replaced by "*it is possible*".
   - Line 140: The sentence "*For the Langevin model including such terms, readers are referred to Paoli and Shariff (2009) and Sardina et al. (2015).*" has been replaced by "*Readers are referred to Paoli and Shariff (2009); Sardina et al. (2015) for the Langevin model including such terms, and also to Chandrakar et al. (2021) which implemented such a Langevin model into the LES Lagrangian cloud model and investigated the impact of entrainment-mixing and turbulent fluctuations on droplet size distributions in a cumulus cloud.*".
   - Line 144: A sentence "*This extension is left for future work.*" has been removed.

We again appreciate the referee's valuable comments which are very constructive to make the paper clearer and better.